# USP7 Deregulation Impairs S Phase Specific DNA Repair after Irradiation in Breast Cancer Cells

**DOI:** 10.3390/biomedicines12040762

**Published:** 2024-03-29

**Authors:** Marie Vogt, Sandra Classen, Ann Kristin Krause, Nadja-Juanita Peter, Cordula Petersen, Kai Rothkamm, Kerstin Borgmann, Felix Meyer

**Affiliations:** Department of Radiotherapy & Radiation Oncology, Hubertus Wald Tumor Center—University Cancer Center Hamburg, University Medical Center Hamburg-Eppendorf, 20246 Hamburg, Germany; mar.vogt@uke.de (M.V.); s.classen@uke.de (S.C.); annbuehrens@hotmail.com (A.K.K.); n.peter@uke.de (N.-J.P.); cor.petersen@uke.de (C.P.); k.rothkamm@uke.de (K.R.); borgmann@uke.de (K.B.)

**Keywords:** USP7, genomic instability, radiosensitization, breast cancer, DNA repair

## Abstract

The ubiquitin specific protease 7 (USP7) is a deubiquitinating enzyme with numerous substrates. Aberrant expression of USP7 is associated with tumor progression. This study aims to investigate how a deregulated USP7 expression affects chromosomal instability and prognosis of breast cancer patients in silico and radiosensitivity and DNA repair in breast cancer cells in vitro. The investigations in silico were performed using overall survival and USP7 mRNA expression data of breast cancer patients. The results showed that a high USP7 expression was associated with increased chromosomal instability and decreased overall survival. The in vitro experiments were performed in a luminal and a triple-negative breast cancer cell line. Proliferation, DNA repair, DNA replication stress, and survival after USP7 overexpression or inhibition and irradiation were analyzed. Both, USP7 inhibition and overexpression resulted in decreased cellular survival, distinct radiosensitization and an increased number of residual DNA double-strand breaks in the S phase following irradiation. RAD51 recruitment and base incorporation were decreased after USP7 inhibition plus irradiation and more single-stranded DNA was detected. The results show that deregulation of USP7 activity disrupts DNA repair in the S phase by increasing DNA replication stress and presents USP7 as a promising target to overcome the radioresistance of breast tumors.

## 1. Introduction

Worldwide, breast cancer is the most common cancer entity in women. In 2020, 2.3 million women were diagnosed with breast cancer and 685,000 women died from the disease globally [1]. Breast cancer is a very heterogeneous disease. The molecular subtype is crucial for the response to therapy. The luminal A subtype is usually defined by a low growth rate and a strong expression of estrogen and progesterone receptors, which are targets for endocrine therapies [2]. Tumors of the luminal B subtype exhibit low expression of the hormone receptors and show increased proliferation due to a high Ki67 expression [3,4]. The Her2-positive subtype does not express estrogen or progesterone receptors but shows a high expression of the growth factor Her2, which offers another therapeutic approach [4,5]. Triple-negative breast cancer (TNBC) shows very low, if any expression of the hormone receptors for estrogen or progesterone and the growth factor Her2. TNBC is characterized by a low 5-year survival rate after therapy, as well as by a high probability of recurrence [6]. For breast cancer therapy, the primary tumor is usually surgically removed and treated with radiotherapy and/or chemotherapy [7]. Despite these therapeutic approaches, the highest proportion of deaths (46.2%) occur within 1 to 5 years after diagnosis [8].

Besides the molecular subtype, the degree of genomic instability of the tumor plays an important role in prognosis [9,10,11]. Genomic instability is mainly driven by defects in DNA repair [12], which lead to the acquisition of new genetic traits in cancer cells, including those necessary to become resistant to therapy. Two of the most well-characterized forms of genomic instability are microsatellite instability (MSI) and chromosome instability (CIN). MSI is characterized by changes in short DNA repeat sequences. CIN results in random irregularities in the distribution of chromosomes during mitosis due to defects in segregation but also due to defects in DNA repair and DNA replication [12]. Thus, CIN describes insertions and deletions, as well as gains and losses of whole chromosomes (aneuploidy) [13,14,15]. To measure CIN, the CIN70 expression signature was derived from a surrogate measure of CIN and is defined as the average expression of 70 genes that correlate with “functional aneuploidy” in solid tumors [16]. Based on this gene signature, a negative prognostic impact of a high CIN70 score on survival was observed for several tumor entities [16]. In breast cancer, a high CIN70 score is associated with a poor prognosis and therapy resistance. It is most pronounced in TNBC, making this subtype difficult to treat [11]. Accordingly, it is of great importance to elucidate the molecular mechanisms of therapy resistance to increase the efficacy of the therapy and prevent recurrence.

### 1.1. The Role of Ubiquitin Specific Protease 7 (USP7) in Cancer

One of the most important posttranslational modifications of proteins is ubiquitination, in which ubiquitin is attached to a protein, and its proteosomal degradation is initiated [17]. This is countered by deubiquitinating enzymes that can degrade ubiquitination and thus protect the substrate from proteosomal degradation. One of the best-studied deubiquitinating enzymes is USP7, playing an essential role in various cellular processes, such as DNA damage recognition and repair, immune response, epigenetics, tumor development and proliferation [18]. Increased USP7 expression has been demonstrated in numerous tumor entities and overexpression of USP7 was shown to promote carcinogenesis and to impair successful therapy [19,20,21,22]. For breast cancer, studies indicate an association between USP7 dysregulation and worse prognosis [23,24]. On a molecular level, the best-described interaction of USP7 is with the transcription factor p53, which regulates the expression of many genes required for DNA repair, cell cycle arrest or apoptosis during genomic stress [25]. Experimental in vitro studies have shown that alterations in USP7 activity influence the stability of various USP7 substrates involved in cell cycle regulation [26,27]. USP7 is also important for accurate and efficient DNA replication through its association with several proteins involved in replication fork progression. Deregulation of USP7 leads to altered numbers of replication origins and thus can affect DNA synthesis rates by stabilizing, e.g., geminin, which inhibits the formation of the prereplicative complex and premature entry into the S phase [28,29]. USP7 has also been shown to stabilize geminin in breast cancer cells. Geminin is important for genomic stability by binding chromatin licensing and DNA replication factor 1 (Cdt1) and preventing the formation of the pre-replicative complex, thereby inhibiting the premature entry of cells into the S phase Thus, USP7 deregulation can lead to DNA re-replication and genomic instability [23,30]. USP7 is directly and indirectly involved in the detection and repair of DNA damage, through the stabilization of various proteins involved. In this context, checkpoint kinase 1 (Chk1), which is activated upon DNA damage and orchestrates cell cycle arrest and homologous recombination (HR) is one of the important substrates of USP7 [31]. Furthermore, USP7 is involved in the repair of both DNA single-strand breaks (SSBs) and DNA double-strand breaks (DSBs) due to the stabilization of several factors involved [26]. With the reduced activity of USP7, it was observed that non-homologous end joining (NHEJ) and HR are impaired, and DNA DSBs are repaired less efficiently. In this regard, USP7 is mainly involved in the stabilization of early DNA repair proteins, such as the mediator of DNA damage checkpoint 1 (MDC1) or the PHD finger protein 8 (PHF8) [32,33].

### 1.2. The Impact of USP7 for Radiation Response

Due to the multiple substrates of USP7 in cell cycle control and DNA damage response, altered expression of USP7 affects cell proliferation and DNA damage repair and thus could influence the response to radiotherapy. The importance of USP7 for the radiation response has been poorly investigated and mainly focused on the effects of USP7 on the transcription factor p53. USP7 downregulation resulted in radioresistance in a *p53* wild-type (wt) human colon carcinoma xenograft model and radiosensitization in *p53* wt laryngeal squamous cell carcinoma cells, which also showed an endogenous increase in USP7 expression after irradiation [34,35]. These contrasting results point to the importance of understanding the underlying mechanisms. In addition to the direct effect of the *p53* status on the role of USP7 in radiosensitivity, USP7 was observed to stabilize the mediator of DNA damage checkpoint protein 1 (MDC1) in cervical cancer cells through deubiquitination. Depletion of USP7 impaired the engagement of the MRE11-RAD50-NBS1 (MRN)-MDC1 complex. Consequently, the recruitment of the downstream factors p53-binding protein 1 (53BP1) and breast cancer protein 1 (BRCA1) to DNA changed and resulted in radiosensitization [32]. This link between USP7 and HR was also observed in human papillomavirus (HPV) positive head and neck squamous cell carcinoma cell lines. It was found that p16 inhibits USP7 due to increased activation of the transcription factor specificity protein 1, and a subsequent degradation of USP7. This resulted in decreased HR and increased sensitivity to irradiation [36].

Overall, studies show an influence of USP7 on the radiation response in cancer cells in both *p53*-dependent and independent mechanisms that directly or indirectly affect HR. It remains unclear whether USP7 also has a radiosensitizing effect in breast cancer tumors, for which a defect in HR has frequently been observed. Therefore, this study investigated the impact of USP7 overexpression and chemical inhibition in combination with irradiation on parameters such as DNA repair and radiosensitization. In parallel, the effects of a deregulated USP7 expression in breast cancer tumors in silico and for the first time, its effects on radiosensitivity in breast cancer cell lines of different subtypes in vitro were analyzed.

## 2. Materials and Methods

### 2.1. Clinical In Silico Analysis

Clinical and mRNA expression data were extracted from the METABRIC dataset, the metastatic breast cancer project dataset and the Cancer Cell Line Encyclopedia dataset, provided by the cBioportal database (http://www.cbioportal.org (accessed on 14 June 2023)). Data from the METABRIC dataset were used for several calculations: (i) Chromosomal instability (Figure 1A): the CIN70 score for each tumor in the METABRIC dataset was calculated according to Birkbak et al. [37] by adding the expression values of all CIN70 genes from 1904 patients. The CIN70 score of the extreme quartiles of USP7 expressing tumors was plotted. (ii) For the overall survival (OS, Figure 1C,D), the survival data of the patients (all patients and the subgroup of patients that received radiotherapy) from the METABRIC dataset were plotted according to the USP7 mRNA expression and the extreme quartiles were plotted and analyzed using a log-rank test. (iii) For the calculation of positive lymph nodes (Appendix A), the number of positive lymph nodes was plotted against the USP7 mRNA expression from the METABRIC dataset. (iv) The USP7 mRNA expression data and DNA copy number alterations (Appendix A) were extracted from the METABRIC database and sorted by molecular subtypes. (v) To calculate the recurrence-free survival (Appendix A) in relation to USP7 mRNA expression, the recurrence-free survival data of the patients from the METABRIC dataset were plotted according to the USP7 mRNA expression and the extreme quartiles were plotted and analyzed using a log-rank test. (vi) Data from the Metastatic Breast Cancer Project dataset were used to analyze a relationship between mutational count and USP7 mRNA expression in breast tumors (Figure 1B) and USP7 mRNA expression in patients with local recurrence (Appendix A). For this, the mutational count data were extracted and plotted against the respective USP7 mRNA expression (Figure 1B), or the USP7 mRNA expression data were allocated to patients with or without local recurrence (Appendix A). (vii) Data from the Cell Line Encyclopedia dataset were used to analyze USP7 expression and mutation count in breast cancer cell lines (Figure 1E and Appendix A). The data were sorted by USP7 mRNA expression in the tumors and the extreme quartiles were plotted.

### 2.2. Cell Culture and Treatments

The cell lines MDA-MB-231 (triple-negative subtype, *p53* mutated) and MCF7 (luminal A subtype, *p53* wildtype) were purchased from the American Type Culture Collection (ATCC, Manassas, VA, USA). Cells were cultivated in DMEM supplemented with 10% FCS and 1% penicillin/streptomycin in incubators at 37 °C, 5% CO_2_ atmosphere and 100% humidity. Cells were irradiated at room temperature with 200 kV and 15 mA, applying a dose rate of 1.2 Gy/min. USP7 was inhibited (P5091, Selleck #S7132, Selleck Chemicals, Houston, TX, USA) with doses ranging from 0.1 to 4 µM for 2–72 h. For measurement of growth rates, 100,000 cells were seeded in T-25 cell culture flasks and treated as indicated. The cell number was measured in duplicates using an automated cell counting system at the indicated time points.

### 2.3. Stable Transfection

MDA-MB-231 cells were transfected with pQFlag-USP7 WT Plasmid (Addgene #46751) using Fugene^®^HD Transfection Reagent (Promega, Madison, WI, USA) according to the manufacturer’s protocol. Successfully transfected cells were selected and grown in DMEM supplemented with 0.1 µg/mL puromycin.

### 2.4. Western Blot

Total protein was extracted from exponentially growing cells. A total of 40 µg of protein was used for SDS-PAGE with a 4–15% gradient gel (Bio-Rad Laboratories, Hercules, CA, USA). After fast transfer (Trans-Blot Turbo Transfer System, Bio-Rad Laboratories) and blocking in 5% bovine serum albumin (BSA) for at least 1 h, proteins were detected using the following primary antibodies: USP7 (Abcam, Cambridge, UK) #ab108931, 1:1000), HSC70 (Santa Cruz, Dallas, TX, USA #sc7298, 1:1000), ATM (Cell Signaling, Danvers, MA, USA #2873, 1:1000) and pATM (Cell Signaling #4526, 1:500). Primary antibodies were marked with IRDYE 680 conjugated anti-mouse IgG, IRDYE 800 conjugated anti-rabbit IgG (LI-COR, Lincoln, NE, USA, 1:7, 500).

### 2.5. Colony Formation Assay

In each well of a 6-well plate, 500 cells were seeded. At 6 h after seeding, the cells were treated with the indicated concentrations of P5091, and at 16 h after seeding, the cells were irradiated. After treatment, cells were cultured for 14 days. Cells were fixed with 70% ethanol and stained with 1% crystal violet (Sigma-Aldrich, Saint Louis, MO, USA). Colonies from more than 50 cells were counted manually and normalized to the corresponding untreated samples.

### 2.6. Immunofluorescence

For foci detection, 100,000–120,000 cells were seeded on culture slides and incubated for 24 h, either untreated or treated with 4 µM P5091. At the indicated time points after irradiation, cells were permeabilized. Cells for PCNA foci were first fixed with ice-cold methanol and after with 4% paraformaldehyde (PFA). All other samples were fixed only with 4% PFA. All samples were blocked for 3 h in 3% BSA. Foci were detected using the following primary antibodies: γH2AX (Ser139, Merck, Darmstadt, Germany #05-636, 1:250), 53BP1 (Novus, Centennial, CO, USA #NB100-305, 1:500), Rad51 (Merck #PC130, 1:500), PCNA (Cell Signaling #13110, 1:500) and RPA (Santa Cruz #sc53496). Following secondary antibodies were used: Alexa Fluor 488 goat anti-rabbit IgG (Cell Signaling, 1:600) and AlexaFluor 647 goat anti-mouse IgG (Cell Signaling, 1:500). Nuclei were stained with DAPI, and the samples were mounted (Vector Laboratories, Newark, CA, USA). Pictures were taken and foci were quantified automatically by the Aklides^®^-system (Medipan, Dahlewitz, Germany). EdU (#C10340, Life Technologies, Carlsbad, CA, USA, 1:1000) was added to the media 30 min before irradiation to the label S phase cells. EdU was stained with 5-FAM-Azide 488 according to the manufacturer’s protocol (#C10340, Life Technologies, 1:250). Fluorescence intensity was quantified using ImageJ software (V. 1.53e). At least 100 cells were analyzed for foci and fluorescence quantification in each condition and experiment.

### 2.7. Cell Cycle Analysis

Exponentially growing cells were harvested and at 24 h after treatment, fixed with ice-cold 80% ethanol. After centrifugation, the cell pellet was resuspended in PBS and stained with DAPI (1:1000) for 1 h in the dark. Flow cytometry analysis was performed using MACSQuant10 with MACSQuantify Software 2.11 (Miltenyi Biotec, Bergisch Gladbach, Germany). The cell cycle distribution was analyzed with ModFit LT™ 3.2 software (Verity Software House, Topsham, ME, USA).

### 2.8. Statistical Analysis

The graph creation, curve fitting, and statistical analysis were performed with GraphPad Prism, Version 9 (GraphPad Software). Data are summarized as mean (+SEM) from at least three independent experiments. Significance was tested by Student’s *t*-test.

## 3. Results

### 3.1. Abnormal USP7 Expression Affects CIN70 Dependent Chromosomal Instability and Overall Survival in Breast Cancer

A high expression of USP7 is associated with poor prognosis in some cancers, including ovarian cancer, prostate cancer, cervical cancer and colorectal cancer (summarized by [22]). For breast cancer, there is some evidence that both low and high USP7 expression correlates with poor breast cancer specific survival in luminal and triple-negative breast cancer [23]. The study results indicated an association between altered USP7 expression and CIN, which is associated with poor prognosis in many tumors, including breast cancer [11,37]. To investigate the relationship between CIN and USP7 expression in more detail, patient data were extracted from the METABRIC database and the quartiles with the highest and lowest USP7 mRNA expression were compared according to the CIN70 score [16]. The CIN70 score was significantly increased in breast tumors with high USP7 mRNA expression in comparison to those with low USP7 expression with 540.7 ± 1.3 and 534.4 ± 1.6, with *p* < 0.01 (Figure 1A). One important cause for an increased CIN70 can be a defect in the DNA damage response or in the DNA repair pathway HR, which would result in an accumulation of mutations. To investigate the relationship between USP7 mRNA expression and mutations in breast tumors, patient data from the Metastatic Breast Cancer Project were extracted and the mutational count of the extreme quartiles of USP7-expressing tumors were compared. The analysis showed that tumors with high mRNA expression of USP7 have a significantly increased mutational count in comparison to USP7 low expressing tumors with 112.4 ± 32.5 and 54.3 ± 5.2, with *p* < 0.0001 (Figure 1B). Lymph node metastases are an important prognostic marker, providing a higher risk for local or distant tumor recurrence. Therefore, the impact of USP7 expression on lymph node metastasis was tested. Patients with high USP7 expression showed a significantly increased number of positive lymph nodes in comparison to patients with USP7 low-expressing tumors (Appendix A). The analyses of recurrence-free survival and USP7 mRNA expression in patients with or without local recurrence showed no significant differences, only a trend for worse recurrence-free survival and an increased USP7 mRNA expression in tumors of patients with local recurrence (Appendix A). Next, the impact of USP7 mRNA expression on the overall survival (OS) of breast cancer patients was analyzed. For this, the upper and lower quartiles of breast tumors with high or low USP7 expression were compared (Figure 1C). A high expression of USP7 showed a significantly decreased overall survival in comparison to patients with breast tumors with low USP7 expression. After radiotherapy, patients with a USP7 high-expressing tumor had a significantly lower overall survival than patients with a USP7 low-expressing tumor, *p* = 0.005 (Figure 1D), comparable to the whole cohort (Figure 1C). This indicates that patients with a USP7 high-expressing tumor do not benefit more from radiotherapy than patients with USP7 low-expressing tumors under standard therapy.

### 3.2. Differences in USP7 Expression among Breast Cancer Subtypes and Cell Lines

Next, it was investigated whether USP7 is differentially expressed in the different breast cancer subtypes (Appendix A). It showed that USP7 mRNA expression is very heterogeneous across all breast cancer subtypes with comparable amounts of normal-, low- and high-expressing tumors.

However, further analysis revealed differences in the *USP7* gene on the DNA level between the breast cancer subtypes: particularly luminal A and B tumors showed an amplification of the *USP7* gene, whereas HER2 positive and TNBC tumors showed no gene amplification of *USP7* but a gain of function of the *USP7* gene, which was lower than in the luminal A and B subtypes (Appendix A). In breast cancer cell lines on the other hand, the analysis of mRNA expression of USP7 from the Cell Line Encyclopedia database showed striking differences within the molecular subtypes (Figure 1E): HER2 positive breast cancer cell lines showed a USP7 expression close to the level of normal tissue with a low variation of −0.4 to +0.02. Large variations in magnitude and direction were observed in TNBC cell lines: the cell lines HCC2157 and HDQ-P1 showed to lowest USP7 expression with −14.3 and −13.8, whereas the highest expressing cell lines were Cal-148 and Du4475 with +21.1 and +13.4, respectively. In luminal A cell lines, USP7 expression was either close to normal tissue or highly overexpressed. The cell lines EFM-19 and HCC1419 showed the highest expression, with +20.6 and +16.0. Cell lines of the luminal B subtype showed a comparable expression pattern, like the luminal A cell lines, e.g., the cell lines MDA-MB-361 and BT-474 showed the highest expression with +21.1 and +12.9, respectively. Further analysis showed that there was a tendency for an increased mutation count in cell lines with high USP7 expression (Appendix A). Overall, the cell lines investigated showed a broad variation in USP7 expression. To further analyze the effect of USP7, cell lines with a low expression, close to the normal tissue level but of different tumor subtypes were used and USP7 protein expression was deregulated artificially to detect alterations depending specifically on the USP7 expression.

### 3.3. USP7 Inhibition and Overexpression Lead to Lower Cellular Survival

To investigate the impact of USP7 overexpression or inhibition, *p53* mutated triple-negative MDA-MB-231 (WT) cells were stably transfected with a USP7 coding plasmid leading to an almost doubled expression of the USP7 protein with 1.88 ± 0.2 to 1.0 ± 0.01, with *p* < 0.001, in several clones which were pooled (USP7 OE cells) (Figure 2A). Surprisingly, these USP7 OE cells showed a significantly decreased plating efficiency (PE) in comparison to MDA-MB-231 WT cells with 0.24 ± 0.02 to 0.32 ± 0.02, with *p* < 0.01 (Figure 2B). For USP7 downregulation, the small molecule inhibitor P5091 was chosen, due to its specificity and the availability of data from other studies (summarized in [38]). After USP7 inhibition by P5091, the PE decreased significantly in both cell lines analyzed. The MDA-MB-231 showed a dose-dependent decrease starting from 0.37 ± 0.04 to 0.19 ± 0.03 with 2 µM of P5091 and 0.04 ± 0.01 with 4 µM and *p* < 0.01 and *p* < 0.0001, respectively (Figure 2C). The *p53 wt* MCF7 showed a decrease starting from 0.39 ± 0.02 to 0.25 ± 0.01 with 2 µM of P5091 and to 0.06 ± 0.01 with 4 µM and *p* < 0.01 and *p* < 0.0001, respectively (Figure 2D). Next, the impact of increasing doses and incubation times of P5091 on cell numbers was investigated. In the MDA-MB-231 cell line (Figure 2E), cell numbers started to decrease significantly at a dose of 4 µM P5091 to a range of 71.07 ± 5.6% to 88.7 ± 3.0% (*p* < 0.05) in comparison to the control for the respective incubation time. After 6 and 8 µM, cell numbers further decreased in a dose-dependent manner. In the MCF7 (Figure 2F), only the highest dose of 8 µM resulted in a significant decrease after 24 h incubation with 62.2 ± 1.7%, with *p* < 0.05 (Figure 2F). After 48 h, significant decreases in proliferation were found at the inhibitor doses of 4–8 µM with 58.4 ± 2.5% to 30 ± 0.7%, with *p* < 0.05 and *p* < 0.01, respectively. In the USP7 OE cells (Figure 2G), USP7 inhibition started to affect cell numbers significantly at a dose of 4 µM, comparable to the MDA-MB-231 WT: With a dose of 4 µM, cell numbers decreased significantly, ranging from 63.6 ± 1.5% to 69.85 ± 4.8%, after 24 and 48 h, respectively, with *p* < 0.05. After 6 µM and 8 µM, proliferation decreased even further, particularly after longer incubation times. Overall, USP7 inhibition led to a dose-dependent decrease in proliferation and clonogenicity in all analyzed cell lines.

### 3.4. USP7 Inhibition and Overexpression Lead to Increased DNA Damage after Irradiation in S Phase

To investigate whether the inhibition of USP7 affects endogenous DNA damage, 53BP1 foci were analyzed in untreated cells, but no significant differences were found. We assumed that a possible effect of USP7 inhibition on DNA repair might emerge if a threshold of DNA damage is exceeded. To investigate this, cells were irradiated with 6 Gy, and early (4 h), as well as residual (24 h) 53BP1 foci, were analyzed (Figure 3A–C). USP7 inhibition resulted in a significant increase in 53BP1 foci in the MDA-MB-231 and the MCF7 cell line after irradiation, whereas it led to a significant decrease in the USP7 OE cell line. After 4 h of irradiation, the number of 53BP1 foci increased in MCF7 cells from 19.7 ± 0.6 after irradiation alone to 30.1 ± 0.8, with *p* < 0.0001, after co-treatment with P5091 and from 12.5 ± 0.5 to 18.4 ± 1.0, with *p* < 0.0001, in the MDA-MB-231 cells. The number decreased significantly in the USP7 OE cells from 13.3 ± 0.7 to 10.4 ± 1.3, with *p* < 0.05. The inhibition of USP7 and irradiation also led to significantly higher numbers of residual 53BP1 foci in comparison to irradiation alone (Figure 3B). The number of residual 53BP1 foci increased from 9.8 ± 0.6 to 16.8 ± 0.6, with *p* < 0.0001, in the MCF7, from 8.9 ± 1.4 to 14.33 ± 0.7, with *p* < 0.001 in the MDA-MB-231 and decreased from 21.1 ± 0.9 to 14.9 ± 2.3, with *p* < 0.05, in the USP7 OE clone. Different from the MCF7 and the MDA-MB-231 WT, the number of residual 53BP1 foci in the USP7 OE cells after irradiation alone increased from 4 h to 24 h and remained significantly higher than in the wild type, with *p* < 0.0001 (Figure 3B,C).

To obtain an indication of which DNA repair pathways are affected, the role of cell cycle phases for DNA damage repair after irradiation and USP7 inhibition was investigated (Figure 3D–F). For this purpose, cells were co-stained for yH2AX foci as a marker of DNA damage, and PCNA, which served as a marker for active DNA replication forks, was used to differentiate between proliferating and non-proliferating cells. Strikingly, the proliferating, PCNA-positive cells showed significantly increased numbers of yH2AX foci after irradiation in combination with USP7 inhibition: in MCF7 the number of yH2AX foci increased from 13.3 ± 1.1 to 20.5 ± 1.7, with *p* < 0.001, and from 12.2 ± 3.0 to 22.2 ± 2.6, with *p* < 0.001, in the MDA-MB-231 cells. The number only decreased slightly in the USP7 OE cells after double-treatment but was significantly increased in comparison to the MDA-MB-231 WT after irradiation alone, with *p* < 0.0001. In PCNA-negative cells, the differences between irradiation alone and co-treatment with the USP7 inhibitor were much smaller and not significant. Overall, the data show that inhibition and overexpression of USP7 both compromise DNA repair of radiation-induced DNA damage, particularly in the S phase.

### 3.5. Altered USP7 Expression Affects RAD51 Foci Formation and Compromises DNA Replication

Radiation induces several kinds of direct and indirect DNA damages. DSB, SSB and base damages (BDs) can all lead to secondary effects, like increased DNA replication stress in the S phase. In the S phase, DSBs are mainly repaired by HR, which also suppresses DNA replication stress by protecting and stabilizing stalled DNA replication forks [39]. An important marker for HR functionality is the ability of cells to form RAD51 foci, which was investigated 6 h after irradiation and USP7 inhibition (Figure 4A,B). USP7 inhibition resulted in a significant decrease in RAD51 foci formation in all cell lines. Even without irradiation, the treatment with P5091 alone reduced the number of RAD51 foci significantly in the MCF7 from 8.2 ± 0.7 to 4.2 ± 0.2, with *p* < 0.0001, and in the MDA-MB-231 WT from 8.4 ± 1.2 to 3.3 ± 0.3, with *p* < 0.0001.

However, the inhibition had no impact on the number of RAD51 foci in the USP7 OE cells. After irradiation, the USP7 inhibitor led to a significant decrease in RAD51 foci from 18.9 ± 0.9 to 12.04 ±0.7, with *p* < 0.0001, compared to irradiation alone in the MCF7cells. In MDA-MB-231 from 13.4 ± 0.7 after only irradiation to 6.6 ± 0.4, with *p* < 0.0001, by co-treatment with P5091 and from 17.9 ± 1.2 to 12.9 ± 0.8, with *p* < 0.001, in the USP7 OE cells. These differences were not due to significant changes in the S phase proportions (Appendix A). To analyze whether the decreased number of RAD51 foci correlated with an increased amount of single-stranded DNA, RPA foci after irradiation were quantified (Figure 4C,D). After irradiation, USP7 inhibition led to a significant increase in RPA foci from 14.6 ± 1.3 to 22.0 ± 1.6, with *p* < 0.001, in MCF7 and from 14.9 ± 0.7 to 30.8 ± 1.4, with *p* < 0.0001, in MDA-MB-231. In the USP7 OE cells, no significant change was found after the P5091 treatment, but the RPA foci number was significantly increased after irradiation alone in comparison to the MDA-MB-231 WT, with *p* < 0.0001. Without irradiation, P5091 alone did not lead to an increase in RPA foci. The increased number of RPA foci after DNA damage and USP7 inhibition indicated a disruption of DNA replication. To further investigate the impact of USP7 inhibition on DNA replication, the number of PCNA foci, which served as a marker for active replication forks, was quantified (Figure 4F). After irradiation, the number of PCNA foci increased in all cell lines. However, the total number of PCNA foci after the combined treatment with P5091 significantly decreased in comparison to irradiation alone. In the MCF7, PCNA foci decreased from 19.7 ± 1.5 after irradiation to only 9.8 ± 2.1, with *p* < 0.001, after combined treatment, and in the MDA-MB-231, they decreased from 23.7 ± 1.1 to 13.2 ± 1.8, with *p* < 0.0001. In the USP7 OE cells, the number of PCNA foci significantly increased after USP7 inhibition from 5.8 ± 0.8 to 10.8 ± 1.2, with a *p* < 0.0001. However, in comparison to the MDA-MB-231 WT, the number of PCNA foci after irradiation alone was significantly lower, with 23.7 ± 1.8 to 5.8 ± 0.8, with *p* < 0.0001. No difference in the number of PCNA foci was found after treatment with P5091 alone in comparison to the control. The significant decrease in active replication forks after irradiation was also reflected by a decreased incorporation of the base analog EdU (Figure 4G,H): in the MDA-MB-231 WT, P5091 alone had no significant impact on EdU incorporation. However, after irradiation, USP7 inhibition led to a significant decrease in incorporated EdU (measured with fluorescence intensity, *p* < 0.0001). In the USP7 OE cells, irradiation alone led to a significant decrease in incorporated EdU, comparable to MDA-MB-231 WT after irradiation plus P5091 treatment (*p* < 0.0001). Overall, USP7 deregulation led to a significant increase in DNA replication stress, leading to an increased amount of ssDNA, lower numbers of active DNA replication forks, decreased base incorporation and in case of USP7 deficiency, disrupted RAD51 accumulation.

### 3.6. USP7 Deregulation Leads to Distinct Radiosensitization

Next, it was analyzed whether the observed effects of USP7 deregulation on DNA repair and replication resulted in radiosensitization of the cell lines. The inhibition of USP7 led to a distinct radiosensitization of all investigated cell lines, independent of the molecular subtype (Figure 5A–D). In MCF7, the D_37_ decreased from 2.6 ± 0.1 to 1.5 ± 0.05 Gy, with significant differences at the dose points 2 and 4 Gy with *p* < 0.01 and *p* < 0.001, respectively. In MDA-MB-231 WT, USP7 inhibition led to a decrease in the D_37_ from 3.9 ± 0.5 to 2.0 ± 0.1 Gy with significant differences at the dose points 2 and 4 Gy with *p* < 0.01 and *p* < 0.001, respectively. USP7 overexpression showed a comparable effect to USP7 inhibition, also leading to a decrease in the D_37_ from 3.9 ± 0.05 to 1.9 ± 0.1 with significant differences at the dose points 2 and 4 Gy, with *p* < 0.01 and *p* < 0.0001, respectively. An additional treatment with P5091 resulted in no colonies after 2 Gy (see Figure 5D). Overall, the data show that USP7 deregulation leads to a distinct radiosensitization, even at lower doses and independent of the p53 status and molecular subtype of the breast cancer cells.

## 4. Discussion

### 4.1. Effect of USP7 Overexpression on Survival and CIN70 of Breast Cancer Patients

Breast cancer is the most common cancer in women and a very heterogeneous disease [40]. According to the WHO, breast cancer caused 685,000 deaths in 2020 alone [1]. There is a substantial need to identify new molecular targets for a more personalized and effective treatment. In this study, we investigated the impacts of an altered USP7 expression in breast tumors on CIN70, OS and OS after radiotherapy. For the analyses, the focus was on the extreme quartiles in terms of USP7 mRNA expression, because (i) its expression in many breast tumors of the METABRIC cohort is comparable to normal tissues (Appendix A) and we assumed that patients with breast tumors, providing an aberrant USP7 mRNA expression, would most likely benefit from an improvement in existing therapies and (ii) when analyzing the entire cohort (i.e., including the majority of patients with normal USP7 expression), there was a risk that small differences between the extreme phenotypes were lost in the large number of patients with tumors with normal USP7 expression. With the expression of USP7, we found a clear association between a high expression of USP7 and an increased CIN70 score in breast tumors, which is known to be associated with poor prognosis [11,16,37]. The significantly decreased OS of breast cancer patients with high USP7 expression is also consistent with observations from another study. The study showed a significant decrease in the survival rate in luminal and TNBC tumors with high or low USP7 expression in comparison to those with median USP7 expression [23]. This study also indicated a correlation between USP7 and CIN, using the threonine tyrosin protein kinase as a marker for CIN. Further studies in other tumor entities were summarized in a systematic review, which included eight studies of five tumor entities and 1192 patients [22]. It showed that a high expression of USP7 is associated with poor prognosis in all included studies and the meta-analysis showed a significantly increased pooled hazard ratio for OS for those patients. Furthermore, a high expression of USP7 also correlated with the presence of lymph node metastases, which is in line with our observations in breast cancer in the in silico analysis (Appendix A) [22]. Chromosomal instability and mutational load result from defects in DNA replication and DNA repair, leading to the assumption that these tumors respond particularly well to therapy [11]. However, it was hypothesized that genomic instability improves the biological fitness of cancer cells from a life-sustaining level to a threshold level. Only outside of this range would cancer cell viability decrease, and an increased therapy response would emerge [37]. Thus, the resulting therapy resistance could be an explanation for why patients with USP7 high-expressing tumors do not benefit from radiotherapy in our analysis. Another factor for genomic instability and therapy resistance could be the basic function of USP7 to stabilize a wide range of proteins, resulting in altered protein levels of, e.g., signaling or DNA repair proteins [32,41]. We could not test this hypothesis because none of the datasets we used for our analyses included protein expression data. Our analysis showed that altered USP7 mRNA expression is not restricted to individual subtypes of breast cancer but that expression is evenly distributed between them (Appendix A). However, our analysis also showed an amplification of the USP7 gene in luminal A and B tumors (Appendix A). USP7 was shown to stabilize the estrogen α receptor and thus has an important function for the biology of these tumors, which is reflected by the in silico analysis, showing an increased copy-number of the *USP7* gene in breast tumors of the luminal A and B subtypes [24].

### 4.2. USP7 Overexpression Enhances Radiosensitivity Due to Increased Replication Stress

The heterogeneous mRNA expression level of USP7 in the molecular breast cancer subtypes (Figure 1E) confirmed our choice to use cell lines with an expression level of USP7 close to the normal tissue of different molecular subtypes and to deregulate the USP7 expression artificially. There is only a very limited number of studies that investigated USP7 overexpression, and the results are contradictory. In this study, we found that the overexpression of USP7 in MDA-MB-231 cells led to a significant decrease in clonogenicity (PE, Figure 2B). This result is contradictory to observations in hepatoblastoma cells, where USP7 overexpression led to a significant increase in clonogenicity in two different cell lines [21]. In colon carcinoma cells, on the other hand, and in line with our observation, USP7 overexpression led to a reduced proliferation, which was attributed to an altered regulation of p53 and Mdm2 [35]. Furthermore, it was hypothesized that p53-independent factors could influence growth delay after USP7 dysregulation [28,42]. Because the USP7 OE cells are based on the *p53* mutated MDA-MB-231 cell line, it can be assumed that the altered stability of other proteins caused the decreased clonogenicity. These very different observations in the studies could be attributed to differences in proliferation, the *p53* status or the tumor entity.

We observed that USP7 overexpression resulted in an increased radiosensitivity (Figure 5C), which is in line with observations in colon carcinoma xenograft models, where USP7 overexpression led to an increased growth inhibition after irradiation [35]. This increased radiosensitivity can be attributed to a significantly increased number of residual DNA double-strand breaks (DSBs) after irradiation (Figure 3C). This is the first study showing that the radiosensitivity of USP7 OE cells was particularly present in the S phase (Figure 3E,F). In the S phase, HR is the most prominent DNA repair pathway for the repair of DSBs and the avoidance of DNA replication stress [11]. HR functionality was not compromised in the USP7 OE cells, as demonstrated by RAD51 foci formation (Figure 4A, [43]). However, after irradiation, the USP7 OE cells provided increased amounts of ssDNA (Figure 4B) in comparison to the MDA-MDA-231 WT cells. Furthermore, the number of DSBs unexpectedly increased from 4 h to 24 h after irradiation (Figure 3B,C). This strongly suggests an accumulation of secondary, replication-stress-associated DSB in these cells due to irradiation-induced DNA replication stress [44,45]. This phenomenon was observed before in wild-type and ATM-deficient fibroblasts, where insufficient phosphorylation of ATM resulted in a replication defect [46]. A deficient phosphorylation of ATM can be ruled out as a cause for the compromised DNA replication as the USP7 OE cells were not deficient in ATM phosphorylation in comparison to the MDA-MB-231 WT (Appendix A). However, we observed that DNA replication was already compromised. The non-irradiated USP7 OE cells had lower numbers of PCNA foci, a decreased base incorporation and an elongated S phase after irradiation (Figure 4F,H and Appendix A). Comparable observations have yet only been made after USP7 depletion [28]. This suggests that both USP7 up- and downregulation influence the tightly regulated balance of proteins needed for accurate DNA replication and thus lead to a compromised DNA replication fork progression and increased susceptibility for replication stress. Our results show that this is particularly apparent after irradiation: the ratio between the number of RPA foci per active replication fork (PCNA foci, Appendix A) was significantly larger in the USP7 OE cells in comparison to the MDA-MB-231 WT cells. Increased replication stress and the resulting accumulation of DNA damage promote genomic instability and clonal evolution in tumors [12,44]. Considering the results in the context of the in silico analysis, it can be assumed that USP7 overexpression does not contribute to improved DNA repair but to chromosomal instability of the tumor by deregulating DNA replication and repair, which in turn contributes to the poor prognosis.

### 4.3. USP7 Inhibition Affects Cell Survival and DNA Replication after Irradiation in Breast Cancer Cells

For the downregulation of USP7, we used the small molecule inhibitor P5091, which is mostly described with effects comparable to a functional knockdown of USP7 with siRNA (summarized in [38]). Treatment with P5091 led to a significant and dose-dependent decrease in PE in the MDA-MB-231 WT and MCF7 cells (Figure 2C,D), which is in line with observations in ovarian cancer cells, lung squamous cell carcinoma cells and colon carcinoma xenograft models after USP7 depletion or inhibition [35,42,47]. This can be attributed to a large increase in G2 phase cells, indicating a G2 arrest (Appendix A), which is confirmed by observations from other studies showing a G2/M cell cycle arrest after USP7 inhibition and a delayed mitotic progression [48,49].

Most strikingly, the inhibition of USP7 resulted in a significant radiosensitization of all investigated cell lines, independent of the USP7 expression level and the molecular subtype (Figure 5A–C). Both, the *p53* mutated MDA-MB-231 and the *p53* WT MCF7 were radiosensitized by USP7 inhibition. An increased cellular radiosensitivity was demonstrated by the increased number of residual DSBs after irradiation in combination with P5091 in comparison to irradiation alone (Figure 3B,C). However, different from the USP7 OE cells, no increase in DSBs between 4 h and 24 h after irradiation was observed, suggesting a different underlying mechanism, like a disruption of DSB repair due to USP7 inhibition. This is in line with observations from others that showed an inhibition of both, HR and NHEJ after USP7 depletion [33,50]. Further investigations showed that USP7 inactivation sensitized the cells, particularly in the S phase for irradiation (Figure 3E). In non-proliferating cells, the effects of USP7 inhibition were much less pronounced (Figure 3F), comparable to the USP7 OE cells. This led to the assumption that these effects were mostly driven by a disruption of HR and less by the NHEJ. Confirming this, treatment with P5091 resulted in a significantly decreased number of RAD51 foci with or without additional irradiation, strongly suggesting impaired HR functionality [43]. This is in line with observations in chronic lymphocytic leukemia (CLL) and cervix carcinoma cells, in which USP7 inhibition resulted in a strong reduction in RAD51 foci formation after irradiation and a general decrease in HR activity, independent of the *p53* status and ATM functionality. In addition to increased radiosensitivity, USP7 inhibition also causes increased sensitivity towards alkylating agents, such as mitomycin C and cyclophosphamide, which require functional HR repair [51]. This was also confirmed for PARP inhibitors, cisplatin and etoposide [52,53,54]. Besides the impaired RAD51 foci formation, the number of RPA foci increased significantly after irradiation, suggesting increased irradiation-induced replication stress, which can mostly be attributed to dysfunctional HR. Mechanistically, studies found that USP7 inhibition disrupts early steps of HR initiation: it was shown that USP7 inhibition destabilizes ring finger protein 168 (RNF168), which compromises the recruitment of BRCA1 [55]. Su and colleagues showed an impaired engagement of the MRN-MDC1 complex and the subsequently compromised recruitment of the downstream factors 53BP1 and BRCA1 to the DNA [32]. Comparable to USP7 overexpression, USP7 inhibition resulted in a generally compromised DNA replication after irradiation with decreased numbers of PCNA foci and lower EdU incorporation (Figure 4C–H), which is in line with observations from Lecona et al. [28]. Overall, the results show that USP7 inhibition results in a significant radiosensitization of proliferating cells due to an efficient disruption of HR repair and amplification of irradiation-induced replication stress in breast cancer cell lines.

In the USP7 OE cells, the effects of USP7 inhibition were overall much less pronounced than in the MDA-MB-231 WT and the MCF7 cells. The most striking effects were a partial rescue of replication processes (Figure 4E,F) but also a decreased accumulation of RAD51 foci (Figure 3A,B) and even further increased radiosensitivity (Figure 5). This indicates the dependency of the cells on a consistent USP7 activity and the resulting steady regulation of the USP7 substrates.

In conclusion, USP7 overexpression in breast tumors is associated with genomic instability and poor survival. This is not necessarily due to therapy resistance, but rather due to the adaption of USP7 overexpressing tumor cells to genomic instability. Due to its potent radiosensitizing effect, USP7 might be a promising target for enhancing therapy efficacy. This effect is independent of the *p53* status. The radiosensitizing effect manifested mostly in the S phase, thus targeting highly proliferating tumor cells. The disruption of HR by USP7 inhibition opens up further treatment options aside from radiation therapy, such as PARP inhibitors and alkylating agents, such as cisplatin. Although our and other studies show promising effects of USP7 inhibition [35,42,56,57], there is currently no clinical phase I study (https://www.cancer.gov/research/participate/clinical-trials-search (accessed on 15 January 2024)) testing it as a therapeutic approach. Furthermore, the experimental in vitro findings reported here require validation in a clinically more relevant in vivo or organoid model and in different molecular subtypes of breast cancer. Therefore, the development of potent and selective USP7 inhibitors still requires intensive research and development efforts before the pre-clinical benefits translate into the clinic.

## Figures and Tables

**Figure 1 biomedicines-12-00762-f001:**
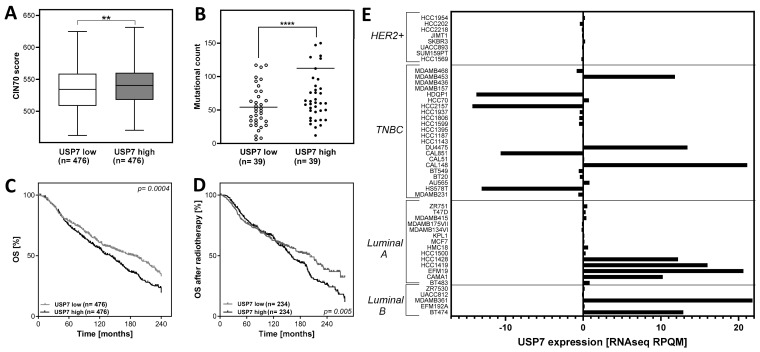
High expression of USP7 correlates with chromosomal instability and survival in breast cancer patients. (**A**): CIN70 score in tumors, sorted by USP7 mRNA expression. Plotted are the extreme quartiles (*n* = 952 out of *n* = 1904). (**B**): Mutational count in patients, sorted by USP7 mRNA expression. Plotted are the extreme quartiles (*n* = 78 out of *n* = 156). (**C**): Kaplan-Meier analysis of USP7 mRNA expression as prognostic factor for overall survival (OS) in breast cancer patients, using the extreme quartiles and analysis with the log-rank test (*n* = 952 out of *n* = 1904) or (**D**): Only breast cancer patients who received radiotherapy (*n* = 568 out of *n* = 1136). (**E**): USP7 mRNA expression in breast cancer cell lines, sorted by molecular subtype. Clinical- and mRNA expression data were extracted from the METABRIC dataset, provided by the cBioportal database (http://www.cbioportal.org (accessed on 14 June 2023)). The CIN70 score was calculated according to Birkbak et al. [37] by adding the expression values of all CIN70 genes from *n* = 1904 patients. For the mutational count, clinical and mRNA expression data were extracted and the extreme quartiles of the number of mutations per tumor were plotted. The USP7 mRNA expression data in cell lines were extracted from the Cancer Cell Line Encyclopedia dataset and sorted by molecular subtype. Asterisks represent significant differences (** *p* < 0.01, **** *p* < 0.0001, Student’s *t*-test).

**Figure 2 biomedicines-12-00762-f002:**
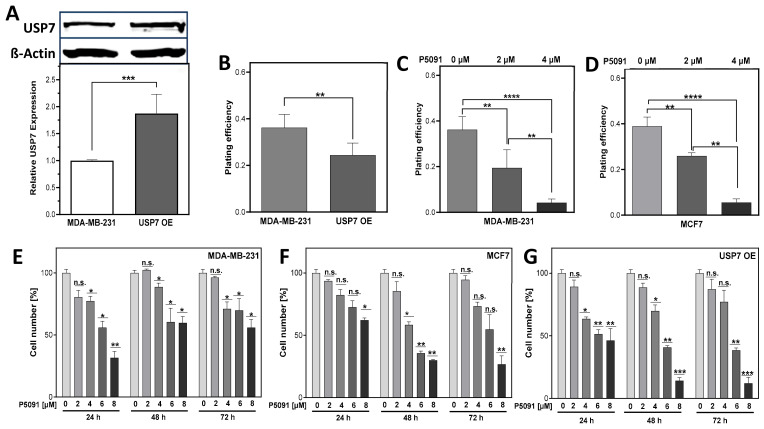
Both, up- and downregulation of USP7 reduces cellular proliferation in breast cancer cells. (**A**): Immunodetection of USP7. MDA-MB-231 cells were stably transfected with pQFlag-USP7 WT plasmid, total protein was extracted from pooled clones. Separated proteins were transferred and detected by appropriate antibodies, with ß-Actin serving as loading control. (**B**–**D**): Plating efficiency of USP7 overexpressing (USP7 OE) or with P5091 inhibited cells. Colonies were stained and counted. (**E**–**G**): Proliferation of cells treated with P5091. Proliferating cells were seeded, treated with indicated doses of P5091 and the cell number was quantified as indicated. Shown are means of three independent experiments ± SEM (n.s. not significant; * *p* < 0.05; ** *p* < 0.01, *** *p* < 0.001, **** *p* < 0.0001, Student’s *t*-test).

**Figure 3 biomedicines-12-00762-f003:**
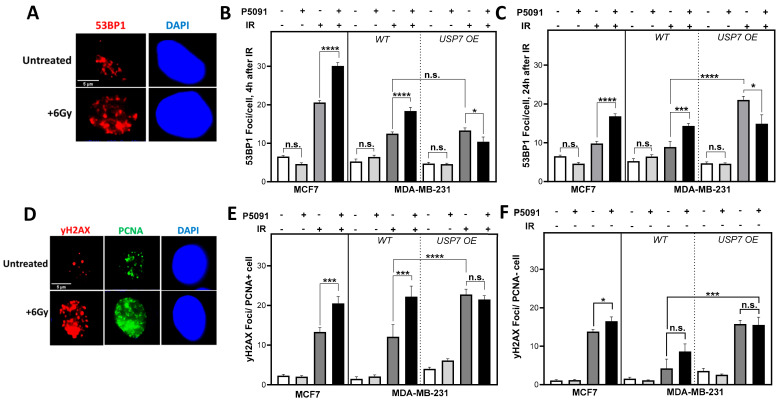
USP7 inhibition and overexpression lead to an increase in DNA damage in the S phase. (**A**): Exemplary depiction of 53BP1 foci 24 h after 6 Gy compared to control. (**B**): Quantification of 53BP1 foci 4 h or (**C**): 24 h after irradiation in the presence or absence of USP7 inhibition. (**D**): Exemplary depiction of yH2AX and PCNA foci 4 h after irradiation or control. (**E**): yH2AX foci in PCNA positive cells or (**F**): PCNA negative cells after irradiation with and without USP7 inhibition. Cells were treated with 4 µm of P5091, irradiated with 6 Gy and immunofluorescence staining was performed 4 h or 24 h after treatment with 53BP1 or 6 h after treatment with PCNA and yH2AX and fluorescent secondary antibodies. Foci were quantified automatically by the Medipan AKLIDES^®^ device. Shown are the means of three independent experiments ± SEM. Asterisks represent significant differences (n.s. not significant; * *p* < 0.05, *** *p* < 0.001, **** *p* < 0.0001, Student’s *t*-test).

**Figure 4 biomedicines-12-00762-f004:**
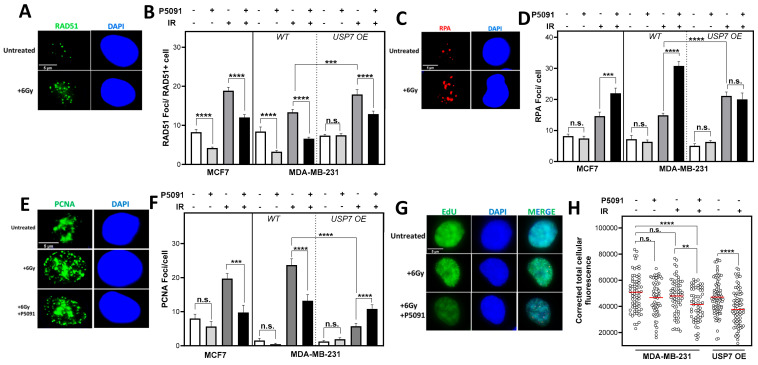
USP7 inhibition and overexpression leads to increased DNA replication stress. (**A**): Examples of RAD51 foci or (**C**): RPA foci or (**E**): PCNA foci after 4/6 h with or without irradiation. (**B**): Quantification of RAD51 foci in RAD51 positive cells, 6 h or (**D**): of RPA-foci 4 h or (**F**): of PCNA foci 4 h after irradiation. Cells were treated with 4 µM P5091 and irradiated with 6 Gy. Fixation and immunofluorescence staining were performed 4/6 h after irradiation with the indicated antibodies plus nuclear staining using DAPI. Foci were quantified automatically by the Medipan AKLIDES^®^ device. (**G**): Exemplary depiction of EdU positive cells after irradiation, with or without USP7 inhibition. (**H**): EdU incorporation after irradiation. Cells were pulse-labeled with EdU and treated with either- or both P5091 and 6 Gy of ionizing radiation. Incorporated EdU was stained and EdU positive cells were captured. Fluorescence intensity was analyzed with the Image J software. Means are depicted in red. Shown are the means of three independent experiments ± SEM. Asterisks represent significant differences (n.s. not significant; ** *p* < 0.01, *** *p* < 0.001, **** *p* < 0.0001, Student’s *t*-test).

**Figure 5 biomedicines-12-00762-f005:**
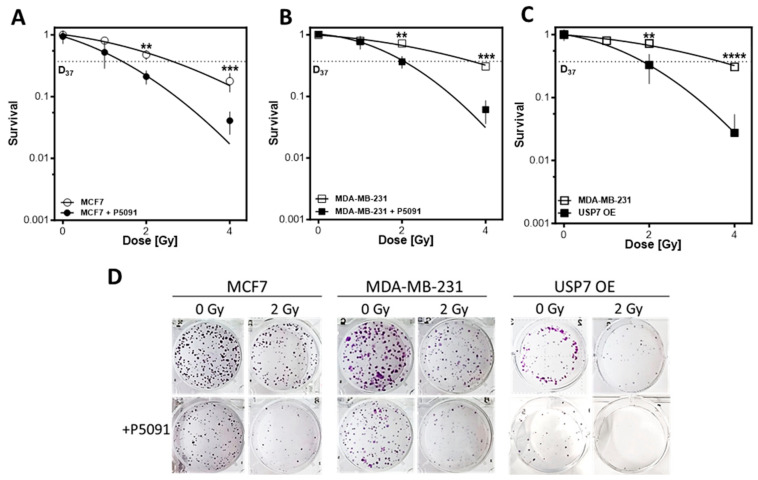
USP7 inhibition and overexpression radiosensitize breast cancer cell lines independent of their p53 status and molecular subtype. (**A**–**C**): Cellular survival after treatment with irradiation alone or in combination with the USP7 inhibitor P5091. Cells were plated, treated with P5091 (2 µM) and/or irradiation, fixed and stained after 14 days and the number of colonies was counted. Shown are the means of three independent experiments ± SEM. Asterisks represent significant differences (** *p* < 0.01, *** *p* < 0.001, **** *p* < 0.0001, Student’s *t*-test). (**D**): Exemplary pictures from colony assays of MCF7, MDA-MB-231 and USP7 OE cells after 2 Gy irradiation, with or without P5091.

## Data Availability

Data are available on request due to restrictions.

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
