# Peer review of "USP7 Deregulation Impairs S Phase Specific DNA Repair after Irradiation in Breast Cancer Cells"

_biomedicines, 2024, doi:10.3390/biomedicines12040762_

Round 1
Reviewer 1 Report
Comments and Suggestions for Authors
The major purpose of the manuscript presented by Vogt et al. was to investigate the impact of deregulated ubiquitin specific protease 7 (USP7) on chromosomal instability and prognosis of breast cancer patients in silico and on DNA repair and survival in breast cancer cells in vitro. In more detail, authors indicate USP7 inhibition and -overexpression to be associated with increased chromosomal instability and lowered overall survival. In addition, both USP7 chemical inhibition and plasmid-mediated overexpression in two breast cancer cells resulted a decreased survival, distinct radiosensitization, and elevated number of residual DNA double-strand breaks and increased detection of single-stranded DNA in S-phase.
In summary, there are no major issues regarding this manuscript as the issue addressed is of high clinical relevance and the authors used a meaningful set of methods and models to corroborate their findings. Overall, the study is well designed and well presented with both in silico analyses and in vitro experiments. There are, however, some minor issues as mentioned successive, aiming to further improve the significance and readability.
Minor points of criticism:
1) Introduction section, especially paragraphs “The role of ubiquitin specific protease 7 (USP7) in cancer” and “The impact of USP7 for radiation response” should be shortened to increase the stringency and readability of the text.
2) In silico analyses, paragraph 3.1: Authors focused on analyses on clinical endpoints overall survival and numbers of positive lymph nodes with the latter to be associated with a risk for local or distant tumor recurrence. Thus, the question arises: Does USP7 deregulation also impact the endpoint local recurrence free survival?
3) Figure 3 A and D may benefit from providing merged pictures including yH2AX, PCNA and DAPI staining. In addition, authors may include a second proof of S-phase specific analyses by exemplarily providing findings on EdU staining.
Comments on the Quality of English Language
Minor editing of English language required. If poosible authors should shorten the text to improve readability and avoid sterotypic sentences like ... lead to/led to.
Reviewer 2 Report
Comments and Suggestions for Authors
The article titled "USP7 deregulation impairs S phase specific DNA repair after irradiation in breast cancer cells" was thoroughly evaluated, and the results indicated that high USP7 expression was associated with an increase in chromosomal instability and a decrease in overall survival. The study included in vitro experiments that were conducted on a luminal and a triple-negative breast cancer cell line. The researchers analyzed proliferation, DNA repair, DNA replication stress, and survival after USP7 overexpression or inhibition and irradiation. The results showed that both USP7 inhibition and overexpression led to decreased cellular survival, distinct radiosensitization, and an increased number of residual DNA double-strand breaks in S phase following irradiation. Furthermore, RAD51 recruitment and base incorporation were decreased after USP7 inhibition plus irradiation, and more single-stranded DNA was detected. The findings indicate that a deregulation of USP7 activity disrupts DNA repair in S phase by increasing DNA replication stress. Therefore, USP7 presents a promising target to overcome radioresistance of breast tumors. However, the authors' should be validated the invitro results by invivo animal studies. Additionally, the authors should be added the bar scale to Fig 3 and Fig 4. Lastly, the authors should include the conclusion/summary of the study to provide a clear understanding of the study's significance and implications.
Comments on the Quality of English LanguageModerate editing of English language required
Reviewer 3 Report
Comments and Suggestions for Authors
In the present manuscript, Vogt and colleagues present their work on radiosensitization by ubiquitin-specific protease 7 (USP7) inhibition or overexpression in mammary carcinoma cells through enhanced replication stress. The authors first confirm the role of USP7 on genomic stability and survival of breast cancer patients by in silico analysis of the METABRIC dataset based on the quartiles with the highest or lowest USP7 mRNA expression. High USP7 gene expression correlates with genomic instability, increased mutation rates, and decreased survival of breast cancer patients, especially in patients with a USP7 high tumor specimen who received radiotherapy. Furthermore, the authors show a wide variation of USP7 gene expression in various breast cancer cell lines and subtypes (Her2+, TNBC, LuminalA/B). However, for their investigations of the role of USP7 in the radiation response of breast cancer cells, the authors do not use a panel of these cell lines but an artificial deregulation in MDA-MB-231 or MCF7 cells for overexpression and inhibition or inhibition only, respectively. Overexpression and targeted inhibition of USP7 in these cell lines lead to reduced plating efficiencies and an impact on the level of radiation-induced DNA double-strand breaks. Using replication markers (PCNA, EdU) and examining resection-dependent DNA double-strand break repair markers (RPA, Rad51), this observation is attributed to increased replicative stress after radiation exposure caused by USP7 deregulation. Finally, the authors demonstrate a general radiosensitization by USP7 overexpression and inhibition based on clonogenic survival. Based on their findings, the authors conclude that USP7 is a relevant clinical target for radiosensitization of tumors or in the context of synthetic lethality with PARP inhibitors.
The authors present an interesting well-thought-out and executed study. However, some comments should be considered in a major revision:
1. Why do the authors focus only on the extreme quartiles of the METABRIC dataset? A general correlation for all participants would be of interest.
2. In Figures 1A and B, if this is retained, the number of patients in each quartile should be given, not the total number/sum of patients.
3. For their in vitro studies, the authors used two cell lines (MDA-MB-231 as TNBC and MCF-7 as Luminal A) that exhibit "normal" USP7 expression. However, overexpression is only investigated in the TNBC cell line MDA-MB-231. Why was it not investigated for the MCF7 cell line, especially since overexpression in luminal A and B is most frequently observed? None of the luminal A/B cell lines show the downregulation observed in TNBC. The inclusion of one or the other cell line with intrinsically increased expression (HCC1428, HCC1419, EFM19, CAMA1, MDAMB361, BT474, CAL148, DU4475, MDAMB453) or downregulation (HS578T, CAL851, HCC2157, HDQP1) of USP7 would also be of interest.
4. Figure 1 should show exemplary images of colony formation corresponding to Figure 5D.
The clonogenic survival data after radiation exposure (section 3.6), which reflect a general radiosensitization of the cell lines by overexpression or inhibition of USP7, should perhaps also be presented here in this context. The authors then turn to a more detailed and mechanistic investigation of these effects.
5. Figure 2: Did the authors investigate the mechanisms of cell death or can they make a statement about the reduced plating efficiency and cellular survival of the cell lines due to overexpression or inhibition of USP7 without exposure to ionizing radiation? There are no differences in the DNA repair assays except for a significantly reduced number of endogenous RAD51 foci due to USP7 inhibition.
If proliferation at 4µM P5091 was already significantly reduced after 24h (Fig. 1E), shouldn't PCNA foci and EdU incorporation also be significantly reduced in the USP7-inhibited samples (Fig. 4 E-H)?
6. Fig. 3A and exemplary images of DNA repair foci in general: the images, which I assume were taken by the Alklides system, do not show the best resolution. After irradiation with 6 Gy, the images shown indicate overlapping of foci and saturation of signals. Why was a relatively high single dose of 6 Gy chosen for these experiments? Were kinetics of dose-response relationships analyzed before the final experiments
7. If the cells are susceptible to replicative stress due to USP7 overexpression or inhibition, this should be tested/confirmed with a potent DNA crosslinker such as mitomycin C or cisplatin. The therapeutic relevance and potential should at least be discussed.
8. Fig. 4E and 4G: An image of untreated cells should be shown.
9. Fig. 4F: In general, PCNA foci serve as markers for DNA damage as well as replication loci. How do the authors differentiate here? The authors should make this clear to the reader.
10. Decreased Rad51 foci (Fig. 4B) and increased RPA foci (Fig. 4D) after radiation exposure suggest an impairment of Rad51 loading at 3' single-strand overhangs at resected DNA double-strand breaks. In the opinion of the authors, what is the underlying cause?
Is an interplay of USP7 with BRCA2, PALB2, or other cofactors for Rad51 loading possible/known and can be discussed?
Reviewer 4 Report
Comments and Suggestions for Authors
Summary of the Manuscript:
Reviewed the manuscript "USP7 deregulation impairs S phase-specific DNA repair after irradiation in breast cancer cells" by Marie Vogt et al. This study investigates the role of USP7 deregulation in breast cancer, focusing on its effects on chromosomal instability, prognosis, radiosensitivity, and DNA repair mechanisms in vitro and in silico. The research demonstrated that high USP7 expression correlates with increased chromosomal instability and reduced overall survival in breast cancer patients. Experimentally, both USP7 inhibition and overexpression in breast cancer cell lines resulted in decreased cellular survival, enhanced radio sensitization, and impaired DNA repair, particularly in the S phase following irradiation.
Strengths:
- The study effectively combines in silico analyses with in vitro experiments, providing a robust examination of USP7's role in breast cancer.
- By linking USP7 expression to patient prognosis and radiosensitivity, the study contributes valuable insights for potential therapeutic targets.
- The investigation into the effects of USP7 on DNA repair mechanisms offers significant molecular insights, particularly concerning RAD51 recruitment and DNA replication stress.
Recommendations:
- Figures 1 and 2 require higher resolution to ensure legibility of legends and details, enhancing the paper's communicative effectiveness.
- Additional studies using a larger variety of breast cancer subtypes and cell lines could be validated and extended these findings.
- Further research into the molecular pathways influenced by USP7, especially in relation to other DNA repair proteins, could provide deeper insights.
Overall Assessment:
The manuscript provides compelling evidence of the critical role of USP7 in breast cancer progression, radiosensitivity, and DNA repair mechanisms. With minor revisions, especially in improving the resolution of figures 1 and 2, to ensure that legends and details are legible, this study stands to significantly contribute to the field. Minor revision are recommended to address these graphical issues alongside the enhancements suggested.
Round 2
Reviewer 2 Report
Comments and Suggestions for Authors
The authors performed significant revisions based on the reviewer's comments. The MS can be considered for publication.
Comments on the Quality of English Language
Minor editing of English language required
Reviewer 3 Report
Comments and Suggestions for Authors
All my comments and concerns have been addressed by the authors. I recommend the acceptance and publication of the manuscript in its current form. I congratulate the authors for their thorough work.